# Emerging Countries and the Effects of the Trade War between US and China

**Monique Carvalho, André Azevedo * and Angélica Massuquetti**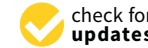

Economics Department, UNISINOS, São Leopoldo 91.330-002, Brazil; monique_fp@yahoo.com.br (M.C.); angelicam@unisinos.br (A.M.)

\* Correspondence: aazevedo@unisinos.br; Tel.: +55-51-35911122

**Abstract:** The aim of the paper is to examine the effects of the US–China trade war on both countries and some emerging economies. Two scenarios are examined, one where only US protectionist measures are considered, and another in which Chinese retaliation is taken into account, using the GTAP (Global Trade Analysis Project) Computable General Equilibrium model. The results showed that, on one hand, the trade war would lead to a reduction in US trade deficit and an increase in domestic production of those sectors affected by higher import tariffs and Chinese producers and consumers would bear the lion's share of the burden of the trade war. But, on the other hand, both countries and the world as a whole would lose in terms of welfare, due to the significant reduction in allocative efficiency, especially in the US, and the loss of terms of trade in the Chinese case. With the increase in protectionism between the two largest global economies, some important emerging countries, not directly involved in the trade war, would benefit by the shift in demand to sectors where they have comparative advantages.

**Keywords:** trade war; import tariffs; retaliation; CGE

**JEL Classification:** F14; C68

## 1. Introduction

Since the beginning of 2018, the multilateral trading system has been challenged by unilateral decisions by the United States of America (US) raising import tariffs for certain trading partners, especially China. The backdrop to these US measures is the increase in the country's trade deficit in recent years, especially with China. According to Comtrade (2018), in 2017, the US trade deficit with China increased to $363 billion, the highest bilateral trade deficit on record. It represents 42% of the total US trade deficit of $861 billion.

US President Donald Trump criticizes his country's huge deficit and attributes it to China's "unfair" trade practices, such as protectionist measures and infringement of intellectual property rights and patents. Therefore, an investigation was opened by the US Trade Representative (USTR), whose final report was delivered on 11 January 2018 to President Trump. Under Section 232 of the Trade Expansion Act of 1962, the agency found that "the quantities and circumstances of steel and aluminum imports threaten to undermine national security as defined in Section 232." The agency suggested import tariffs of 24% on all steel products in all countries and 7.7% on all aluminum products in all countries (USTR 2018).[1]

---

[1]  The USTR report (USTR 2018) found that the current quantities of steel and aluminum imports in the circumstances of global overcapacity of these products would be "weakening the domestic economy", resulting in the persistent threat of

Following the recommendations of the USTR, President Donald Trump signed on 8 March 2018 a regulation imposing an additional 25% ad valorem duty on steel imports and 10% on aluminum at all countries. In addition, Trump announced in April 2018 a list of Chinese products that would suffer a surcharge on imports, equivalent to $50 billion. In the same month, China notified the World Trade Organization (WTO) of its retaliatory measures to the US by presenting a list of products on which import tariffs will also be imposed, including the imposition of a 25% tariff on imported soybeans from the USA.[2]

There have so far been a small number of publications analyzing the impacts of the US–China trade war because these trade policies are fairly recent. However, some articles, using the computable general equilibrium model, have already demonstrated the harmful effects of increased protectionism mainly on US and China, especially in terms of trade and welfare reduction, such as Ciuriak and Xiao (2018) and Bollen and Rojas-Romagosa (2018). These authors focused on economic effects in countries directly involved in the trade war and other potential targets, generally developed countries, such as the European Union, or countries geographically close to the US (Canada and Mexico), with emphasis on those sectors that were initially affected by the measures (steel and aluminum).

In this sense, the two major contributions of this article are to examine the impacts of protectionist measures, including developing countries such as Argentina, Brazil, and India, and covering other sectors of special interest to this group of countries, such as soy and milk. With the surge in protectionism between the two largest global economies, it is natural that emerging countries not directly involved in the trade war can benefit by the shift in demand to sectors where they have comparative advantages. Therefore, even if the trade dispute generates losses, in terms of welfare and trade, for the US and China and for the world as a whole, certain sectors of emerging countries can benefit. Two scenarios are examined: one where only US measures are considered and another in which Chinese retaliation is taken into account.

Therefore, the objective of this study is to estimate the economic effects caused by this trade conflict using a computable general equilibrium model, based on the version 9 of the Global Trade Analysis Project (GTAP) database. Besides examining the impacts on trade balances, production and welfare caused by the US–China trade war on countries directly involved in the conflict, the sector aggregation employed in the paper allow for an analysis of the sectors that would benefit most emerging countries. The first scenario corresponds to the unilateral imposition of US import tariffs on products from China and other countries, while the second considers the Chinese retaliation against the US.

The results indicate that, while there would be an increase in production and a reduction of the US trade deficit in both scenarios, something that President Trump pursued, the greater protectionism would lead to a loss of welfare due to a worse allocation of productive resources in the US economy, which would be even higher in China. For emerging countries that were not directly affected by the measures of the trade war, there would be gains in terms of exports and welfare, especially in sectors where these countries are competitive.

The remainder of the paper is organized as follows. The second section presents the methodology and description of the simulations and the protectionist measures adopted by the US and China. The third section analyzes the results of the two scenarios, with emphasis on international trade and welfare of the regions involved in the simulations. The fourth section compares the results with the still incipient literature dealing with the theme. Finally, the final section concludes.

---

　　further closures of production facilities of domestic steel and aluminum and thereby reducing the ability to meet national security production requirements on a national emergency scale.

[2]　　In June 2018, the EU followed the same path and reported its retaliatory measures to the WTO.

## 2. Methodology

The paper employs a computable general equilibrium model, using the 9th version of the standard GTAP database, based on perfect competition and constant returns to scale. With initial equilibrium in 2011, the database covers 57 productive sectors and 140 regions in the world.

### 2.1. GTAP Overview

This model was developed to determine the impact of trade flows in different sectors and regions of the world, generating results of global consistency (Hertel 1997). The GTAP is composed of equations based on microeconomic fundamentals that portray the behavior of families and firms belonging to each of the modeled regions, as well as interregional flows, considering global transportation costs, with a typically neoclassical closure. The model uses a three-level structure in the specification of the production function: at the first level, the production function assumes zero substitutability between primary production factors and intermediate inputs (Leontief technology). As a result, the optimal mix of primary factors is independent of prices of intermediate inputs, while the optimal mix of intermediate inputs is invariant with respect to price of primary factors[3]; at the second level, it involves a constant elasticity of substitution between inputs and between factors of production. Imported intermediates are assumed to be separable from domestically produced intermediate inputs, that is to say that firms first determine the optimal mix of domestic and imported goods and only then decide the sourcing of their imports (Armington assumption); and at the third level, a constant substitution elasticity is assumed between inputs imported from different origins (Hertel 1997).[4]

The macroeconomic closure of the model is short-term and incorporates the law of constant returns to scale. The investment rate is determined by savings. Prices and quantities of commodities are considered endogenous. Stocks of land, capital and labor and the variables linked to technological change are exogenous. Modeling the maximizing behavior of utility and profits of economic agents, allows the estimation of welfare changes. In order to solve the model, we used the numerical method of Gragg, which reduces possible distortions contained in the linear methods of Johansen and Euler, which allows one to specify a greater number of steps, offering an accurate system solution (Hertel 1997).

One of the advantages of general equilibrium models, such as the GTAP, is the possibility of simulating the impacts on welfare. The GTAP also allows for the decomposition of the welfare effects: allocative efficiency, terms of trade and the investment-saving component (I-S). This calculation is associated with equivalent variations of regional and world income, through the income that would be required to reach a certain level of utility. Allocation effects show that a share of regional income from efficiency gains (or losses) is caused by the removal (inclusion) of distortions caused by the incidence of tariffs on trade. Thus, for example, cheaper imported products bring about gains both through increased consumption and in the way domestic productive resources are used. The terms of trade are affected by the variation in export prices related to the cut or increase in tariffs. The I-S is a function of saving and investment prices and the situation as a given region appears in the net saving balance (Monte and Teixeira 2007).[5]

---

[3] Primary factors of production comprise capital, skilled and non-skilled labor, which are mobile between commodity groups that maintain invariant prices. Land and natural resources are sluggish. The degree of factor mobility is governed by a constant elasticity of transformation.

[4] The values of the elasticity of substitution between primary factors (ESUBVA), between domestic and imported goods in the Armington aggregation structure (ESUBD) and between imports from different sources (ESUBM) are provided in Appendix A.

[5] The model also includes a global bank that intermediates between global savings and regional investments, selling saving goods to each regional household to satisfy their demand for savings and buying shares in a portfolio of regional investment (Hertel 1997). Savings is an argument in regional household utility function and constrained optimization leads to a demand for homogeneous saving goods, which as any other good depends on income of the household and its relative price. Once the global bank assembled all regional savings, there are two approaches by which the global bank can allocate regional investments. The first, so-called 'fixed regional composition' (which is used in all simulations in this paper), assumes that regional composition of global capital stocks is left unaltered in the simulation. Therefore, regional and global investments

The computable general equilibrium model developed by GTAP was adopted in this study to assess the potential impact of the US–China tariff war. The model is one of the most popular analytical tools for assessing the economic effects of free trade agreements as well as the effects of trade wars. Compared with a simple equation econometric model or the partial equilibrium analysis method, the model has the advantage of capturing the input–output relationship between industry and other sectors in the open global economy scenario and thus improves the robustness of the results of the estimates for the market (Hertel 1997).

In order to analyze the effects of a trade war, shocks are carried out on tms, which corresponds to the import tariff of the sector *i* imposed on the exports of country r by the country *s*, in percentage variation. The increase in tms causes an increase in the variable pms(*i*,*r*,*s*), the import price of product *i* provided by country *r* to region *s*. The pms is obtained through tms and pcif, which is the CIF world price of sector *i* supplied from *r* to *s*, as shown in Equation (1).[6]

$$\text{pms}_{(i,r,s)} = \text{tms}_{(i,r,s)} + \text{pcif}_{(i,r,s)} \tag{1}$$

The increase in domestic price of imports of sector *i* from a specific trading partner has two direct effects. First, it causes a rise in aggregate prices of imports of sector *i*, making imported products relatively more expensive, the so-called pim(*i*,*s*). The variable pim is obtained from MSHRS, which denotes the participation of each region in the imports of sector *i* in the country s (in%), and the pms, according to Equation (2).[7]

$$\text{pim}_{(i,s)} = \sum^{r} \text{MSHRS}_{(i,r,s)} \times \text{pms}_{(i,r,s)} \text{pim}_{(i,r,s)} \tag{2}$$

The second effect is to reduce the imports of the regions that have suffered tariff increases in favor of the others that were not affected by the protectionist measures, called qxs(*i*,*r*,*s*), as shown in Equation (3). The variable is obtained from qim, which are the aggregate imports of sector *i* of the country *s*; esubm, which is the elasticity of substitution between imports and domestic products *i* in region *s*; and the difference between pms and pim.

$$\text{qxs}_{(i,r,s)} = \text{qim}_{(i,s)} - \text{esubm}_{(i)} \times \left( \text{pms}_{(i,r,s)} - \text{pim}_{(i,s)} \right) \tag{3}$$

Finally, demand will then be directed to domestic goods, leading to an increase in production in region, according to Equation (4).[8]

$$\text{q}_{\text{o}(s)} = \text{SHRDM}_{(i,s)} \times \text{qds}_{(i,s)} + \text{SHRST}_{(i,s)} \times \text{qst}_{(i,s)} + \sum^{s} \text{SHRXMD}_{(i,r,s)} \times \text{qxs}_{(i,r,s)} \tag{4}$$

When using the model to assess the economic impact of a trade policy within an open economy with many trading partners and many sectors, a new value is assigned corresponding to the exogenous variable that represents the tariff shock (tms). Specifically, the import tariff was increased on the

---

move together and the rates of return in each region will differ. The second mechanism (rate of return component) is an alternative investment approach, in which the rates of return are the same in all regions. Investment depends on expected rate of return in the next period, which declines as capital stock increases.

[6] The equations were obtained from AnalyseGE, which uses TABmate and ViewHAR written by Mark Horridge and is available from the RunGTAP software.

[7] The higher is the share of region in the countries imports of product *i*, the higher would be the impact on aggregate prices of imports of this product.

[8] Where qo is output of sector *i* in country *s* (% change); SHRDM is the share of domestic sales of sector *i* in the country *s*; qds is the domestic sales of *i* in region *s*; SHRST is the share of sales of *i* to global transportation services in *s*; qst is the sales of sector *i* to international transport sector; SHRXMD is the share of export sales of product *i* provided by country *r* to region *s*; and qxs are the exports of *i* from country r to region *s* (% variation).

products listed by the countries involved in the trade war, starting from the tariff practiced at the initial equilibrium plus the additional tariff determined by those countries.[9]

### 2.2. Regional and Sector Aggregation and Simulations

The regional aggregation of the study comprises ten regions that consider the world's largest producers of steel, aluminum, and soybeans: USA, China, Brazil, Argentina, India, Canada, Russia, Mexico, and the EU, and aggregates the remaining countries in "Others". As for the sectoral aggregation, it comprises 18 sectors, including those who suffered the tariff shocks both from the US and China in the trade war between them. The sectors that will not suffer shocks were organized in "other industrialized", "other not industrialized", and "services" (Table 1).

**Table 1.** Regional and sector aggregation.

| Regional Aggregation | Sectoral Aggregation |
|---|---|
| China | Iron & Steel |
| US | Aluminum |
| Brazil | Soybeans |
| Argentina | Primary products |
| India | Other not industrialized |
| Canada | Other industrialized |
| Russia | Dairy products |
| Mexico | Processed Rice |
| EU | Other Food |
| Other Countries | Beverages & Tobacco |
| | Petroleum & Coke |
| | Chemicals |
| | Motor vehicles and parts |
| | Other Transport Equipment |
| | Electronic Equipment |
| | Other Machinery |
| | Other Manufacturing |
| | Services |

Source: Prepared by the authors.

Based on the measures taken by the US in the first stage of the trade war, the paper established the imposition of tariffs on steel and aluminum globally. However, some partners were not affected by the measures, including Brazil (it was decided to establish a maximum export quota), Argentina, Mexico, and Canada, the latter two due to the renegotiation of NAFTA.

The US released a list of 818 Chinese products on which 25% of additional import tariffs would be levied (USTR 2018). The charge on this list came into effect on 6 July 2018, totaling an equivalent of $34 billion in Chinese imports. These products include passenger vehicles, radio transmitters, aircraft parts, hard disks, medical and precision instruments, tires, nuclear reactors and boilers. In response, China retaliated with a list of 545 US products totaling $34 billion in imports (MOFCOM 2018). The list contains agricultural products, food, and vehicles. In this study, these 1363 SH6 codes listed by the US and China were distributed according to the Organization for Economic Cooperation and Development (OECD) classification (Primary, Low Technology, Medium–Low Technology, Medium–High Technology, High Technology, and Services) and its corresponding sectors in the GTAP sectors).[10] Table 2 summarizes the main tariff measures adopted by the USA and China, which are the subject of this study.

---

[9]　The economic impact of the tariff shock is reflected by the change in value of the endogenous variables pms, qxs, qo, qds and pim, comparing its initial value and that obtained in the new equilibrium after the simulation.

[10]　The detailed sectoral aggregation, according to the GTAP sectors, and their corresponding OECD classification, is given in Appendices B and C, respectively.

**Table 2.** Resume of Measures adopted by the US and China.

| Products | Country Adopting the Measures | Ad Valorem Import Tariff | Countries Targeted |
|---|---|---|---|
| Iron & Steel | US | 25% | China, India, Russia, EU and other countries |
| Aluminum | US | 10% | China, India, Russia, EU and other countries |
| US list with 818 Chinese products | US | 25% | China |
| list of China with 818 Chinese products | China | 25% | US |

Source: Prepared by the authors.

Two scenarios were created to examine the effects of the trade war. The first scenario portrays the effects of US unilateral protectionist measures, while the second scenario portrays the impact of Chinese retaliation.

Scenario 1: corresponds to the unilateral imposition of US tariffs on:

(a)   US additional 25% import duty on steel from China, India, Russia, EU and other countries;

(b)   Additional 10% US import tariff on aluminum from China, India, Russia, EU and other countries;

(c)   An additional 25% charge on Chinese products listed by the US.

Scenario 2: Chinese retaliation, with the imposition of:

(a)   An additional 25% tariff on the US products listed by China.

US protectionism and Chinese retaliation may affect exports of relevant products on the export agenda of emerging countries. In 2017, China imported 60% of all soybeans traded on the world market, and Brazil and Argentina, for example, are two of the largest suppliers of the product to the Asian market. Under this scenario of a trade war between the world's largest exporters, this study seeks to identify its effects on key sectors of the emerging economy using a computable general equilibrium model.

## 3. Results

This section analyzes the changes in production, exports, imports, trade balance, and welfare resulting from the US–China trade war, for each scenario of tariff imposition: first, the US unilaterally charges various countries, especially China; and in the second, China retaliates against the US.[11]

### 3.1. Production

Table 3 shows the effects of the shocks on the production of the countries. In that first simulation, in which only the US imposed their tariffs, it is possible to notice that there would be a greater increase in production of iron & steel (5.71%), electronic equipment (5.78%), and aluminum (2.88%) in the US, although it would also occur to a lesser extent in other sectors, with the exception of motor vehicles and parts (−0.47%) and other transport equipment (−1.29%). In China, there would be a significant reduction in production only for part of the targeted sectors, such as other manufacturing (−4.07%) and electronic equipment (−8.09%), not having strong effects on production of others. There would be iron & steel and aluminum production gains in emerging countries, which would not be subject to the application of the US Section 232 import tariff. Iron & steel production would increase 2.49% in Brazil, 2.91% in Argentina, and 1.5% in Mexico, while aluminum production would increase by 1% in Brazil, 3.58% in Argentina, and 4.22% in Mexico.

---

[11]   The results were obtained from RunGTAP, which allows the user to run simulations interactively in a Windows environment using the GTAP general equilibrium model.

**Table 3.** Variation of domestic production (%).

| Sectors | China | US | Brazil | Argentina | India | Canada | Russia | Mexico | EU | Other |
|---|---|---|---|---|---|---|---|---|---|---|
| | | | | **Scenario 1** | | | | | | |
| Iron & Steel | 0.73 | 5.71 | 2.49 | 2.91 | −1.27 | 11.20 | −1.78 | 1.50 | −1.68 | −2.02 |
| Aluminum | 1.11 | 2.88 | 1.00 | 3.58 | −1.24 | 9.26 | −2.36 | 4.22 | −1.32 | −2.10 |
| Soybeans | 1.91 | 0.04 | −0.47 | −0.36 | −0.01 | −1.74 | 0.08 | −2.48 | 0.05 | −0.03 |
| Primary products | 0.62 | 0.04 | −0.35 | −0.46 | −0.04 | −1.49 | 0.09 | −1.51 | 0.00 | −0.06 |
| Other not industrialized | 1.64 | 0.01 | −0.53 | −0.26 | −0.18 | −0.56 | 0.01 | −2.03 | −0.15 | −0.12 |
| Other industrialized | 2.49 | −0.54 | −0.68 | −0.57 | −0.65 | −2.39 | −0.49 | −4.94 | −0.72 | −0.91 |
| Dairy products | 0.03 | 0.01 | 0.09 | −0.13 | 0.07 | −0.03 | 0.09 | −0.52 | 0.05 | −0.02 |
| Processed Rice | 0.36 | −0.07 | −0.06 | −0.26 | 0.04 | −0.09 | 0.15 | −0.68 | 0.02 | 0.03 |
| Other Food | 0.55 | 0.00 | −0.15 | −0.25 | 0.01 | −0.97 | −0.04 | −0.84 | −0.04 | −0.10 |
| Beverages & Tobacco | −0.13 | 0.00 | 0.03 | 0.02 | 0.06 | −0.13 | 0.07 | −0.42 | 0.03 | 0.05 |
| Petroleum & Coke | 0.15 | 0.10 | 0.02 | −0.06 | −0.03 | −0.05 | 0.11 | −1.12 | 0.05 | −0.08 |
| Chemicals | 0.38 | 0.84 | −0.41 | −0.71 | −0.11 | −0.24 | 0.18 | −3.92 | 0.17 | −0.35 |
| Motor vehicles and parts | −0.11 | −0.47 | 0.24 | −0.13 | 0.16 | −0.09 | 0.30 | −4.98 | 0.20 | 0.27 |
| Other Transport Equipment | 1.98 | −1.29 | −0.83 | −0.37 | −0.26 | −3.29 | 0.55 | −4.94 | 0.07 | −0.34 |
| Electronic Equipment | −8.09 | 5.78 | −0.19 | −1.30 | −0.26 | 7.66 | −1.05 | 17.78 | −0.38 | 2.63 |
| Other Machinery | −0.61 | 0.93 | −0.40 | −1.32 | −0.16 | 1.79 | 0.06 | −0.88 | 0.22 | −0.07 |
| Other Manufacturing | −4.07 | 4.11 | 0.23 | 0.05 | 2.54 | 1.08 | 0.17 | 0.46 | 0.59 | 1.86 |
| Services | −0.21 | −0.12 | 0.11 | 0.10 | 0.04 | 0.02 | 0.07 | 0.16 | 0.06 | 0.07 |
| | | | | **Scenario 2** | | | | | | |
| Iron & Steel | 0.73 | 6.00 | 1.90 | 2.14 | −1.29 | 11.12 | −1.83 | 1.49 | −1.68 | −2.06 |
| Aluminum | 1.12 | 3.40 | −0.28 | 2.17 | −1.28 | 9.26 | −2.48 | 4.31 | −1.36 | −2.17 |
| Soybeans | 6.43 | −13.92 | 9.30 | 4.47 | 0.01 | −0.06 | 0.16 | −4.15 | 0.09 | 0.24 |
| Primary products | 0.81 | −0.21 | −0.95 | −2.13 | 0.03 | −1.58 | 0.11 | −1.63 | 0.09 | 0.02 |
| Other not industrialized | 1.61 | 0.18 | −1.15 | −0.72 | −0.21 | −0.57 | 0.01 | −1.99 | −0.15 | −0.12 |
| Other industrialized | 2.39 | −0.40 | −0.98 | −0.80 | −0.66 | −2.45 | −0.49 | −4.95 | −0.71 | −0.91 |
| Dairy products | 0.36 | −0.27 | 0.13 | −0.24 | 0.06 | −0.03 | 0.09 | −0.52 | 0.06 | 0.04 |
| Processed Rice | 0.28 | 0.19 | −0.14 | −1.16 | 0.04 | 0.04 | 0.18 | −0.68 | 0.03 | 0.04 |
| Other Food | 0.52 | −0.15 | −0.30 | −0.54 | 0.01 | −0.96 | 0.13 | −0.80 | −0.01 | −0.02 |
| Beverages & Tobacco | −0.13 | −0.12 | −0.01 | −0.02 | 0.06 | −0.13 | 0.07 | −0.41 | 0.05 | 0.06 |
| Petroleum & Coke | 0.12 | 0.11 | −0.06 | −0.05 | −0.04 | −0.05 | 0.09 | −1.12 | 0.04 | −0.10 |
| Chemicals | 0.32 | 1.09 | −0.61 | −1.02 | −0.14 | −0.33 | 0.13 | −3.94 | 0.12 | −0.40 |
| Motor vehicles and parts | 0.38 | −1.13 | 0.15 | −0.51 | 0.18 | −0.26 | 0.29 | −5.04 | 0.37 | 0.40 |
| Other Transport Equipment | 1.92 | −0.89 | −1.57 | −0.64 | −0.31 | −3.38 | 0.42 | −4.96 | −0.06 | −0.46 |
| Electronic Equipment | −8.06 | 6.14 | −0.26 | −1.69 | −0.32 | 7.56 | −1.13 | 17.71 | −0.45 | 2.50 |
| Other Machinery | −0.62 | 1.26 | −1.03 | −2.06 | −0.20 | 1.60 | 0.02 | −1.01 | 0.15 | −0.19 |
| Other Manufacturing | −4.11 | 4.41 | 0.24 | 0.07 | 2.47 | 1.06 | 0.14 | 0.46 | 0.55 | 1.81 |
| Services | −0.26 | −0.12 | 0.12 | 0.12 | 0.03 | 0.04 | 0.08 | 0.17 | 0.06 | 0.07 |

Source: Author's calculation.

Even with Chinese retaliation, its production of high-technological sectors would continue to fall by similar amounts, given the maintenance of US tariffs on the country's exports. The increase in iron & steel and aluminum production would remain practically at the same levels for Canada and Mexico, while in Argentina and Brazil, steel production would increase to a lesser extent and aluminum production would fall in Brazil. With regard to the sectors protected by China, there would be a decrease in Chinese production only for beverages & tobacco (0.13%), but in the others there would be an increase in production, especially soybeans (6.43%). As for US production in the second scenario, there would be a reduction in all sectors targeted by Chinese tariffs, with the exception of processed rice (0.19%).

The most affected sector by the Chinese retaliation in the US would be soybeans, with a 13.92% reduction in production. As China diverts its import demand of soybeans from US (the target of China's tariffs) to its Latin American trading partners, there would be an increase in Brazilian production by 9.3%, followed by 4.47% in Argentina. In both countries, this increase in soybeans production would occur to the detriment of other primary products, indicating a replacement of crops which are also important in the countries' export agenda. According to Table 3, US production of iron & steel, aluminum, and other protected sectors, in general, would increase, as indicated by Equation (4) of Section 2. In China, the high-technology sectors would be the most affected. In most emerging countries, there would be an increase in production of iron & steel and soybeans benefited by the trade war.

### 3.2. Imports, Exports, and Trade Balance

This section examines the effects of simulations on domestic prices of goods supplied by the trading partner, exports by the trading partner, the market price of aggregate imports and the volume of aggregate imports.

Table 4 shows the effects of the two scenarios for the USA. The new import tariffs for Chinese products are listed in the first column of the table, i.e., with the ad valorem tariff increase on iron & steel (25%) and aluminum (10%). As a consequence, there would be an increase in the prices of these products imported by the US from China (as predicted by Equation (1)). It can be noted in Scenario 1 that increasing the steel tariff by 25% would result in the Chinese iron & steel price rising by 23.23%. As a result, there would be a reduction in US imports of this sector from China by 53.62% (see Equation (3)). The increase in domestic import prices from China also causes a rise of 13.21% in the US sector aggregate prices of imports (denoted by Equation (2)). Given the significant drop in imports from China's steel sector, there would be a 23.48% reduction in US aggregate iron & steel imports, benefiting domestic production (as described in Equation (4)). The other sectors affected by higher tariffs would suffer similar consequences. It is possible to note a significant reduction in the US imports from China in those sectors affected by US protectionist measures in both scenarios. The sector less affected would be aluminum, but even so there would be a reduction in US imports from China of 30.02% in scenario 1 and 29.97% in scenario 2.

**Table 4.** US trade data (%).

| Sectors | US Import Tariffs on China's Exports (%) | US Import Prices of Product *i* from China | US imports from China | US Aggregate Prices of Imports | US Aggregate Imports |
|---|---|---|---|---|---|
| | | **Scenario 1** | | | |
| Iron & Steel | 26.03 | 23.23 | −53.62 | 13.21 | −23.48 |
| Aluminum | 13.19 | 8.36 | −30.02 | 4.83 | −7.55 |
| Soybeans | 0.01 | −0.99 | 6.67 | 0.40 | −0.36 |
| Primary products | 1.12 | −1.38 | 10.50 | 0.40 | −0.55 |
| Other not industrialized | 0.31 | −0.36 | 6.41 | 0.16 | 0.22 |
| Other industrialized | 7.57 | −1.51 | 10.28 | −0.36 | 1.61 |
| Dairy products | 5.94 | −1.15 | 10.49 | 0.26 | −0.35 |
| Processed Rice | 4.36 | −1.42 | 9.00 | 0.20 | 0.15 |
| Other Food | 2.76 | −1.48 | 7.07 | 0.29 | −0.30 |
| Beverages & Tobacco | 4.10 | −1.72 | 4.86 | 0.49 | −0.38 |
| Petroleum & Coke | 25.17 | 24.47 | −59.52 | 0.43 | −0.30 |
| Chemicals | 27.75 | 22.56 | −71.56 | 1.85 | −3.55 |
| Motor vehicles and parts | 25.86 | 22.94 | −67.04 | 1.26 | −2.31 |
| Other Transport Equipment | 28.43 | 22.25 | −81.09 | 1.11 | −3.20 |
| Electronic Equipment | 25.25 | 23.10 | −75.05 | 6.78 | −12.78 |
| Other Machinery | 26.47 | 22.76 | −77.31 | 3.27 | −7.99 |
| Other Manufacturing | 26.52 | 22.50 | −70.06 | 5.93 | −10.98 |
| Services | 0.00 | −1.97 | 8.79 | 0.25 | −0.24 |
| | | **Scenario 2** | | | |
| Iron & Steel | 26.03 | 23.25 | −53.59 | 13.28 | −23.67 |
| Aluminum | 13.19 | 8.37 | −29.97 | 4.86 | −7.59 |
| Soybeans | 0.01 | 0.00 | −4.37 | 0.62 | −7.21 |
| Primary products | 1.12 | −0.96 | 6.50 | 0.46 | −2.07 |
| Other not industrialized | 0.31 | −0.33 | 6.16 | 0.19 | 0.05 |
| Other industrialized | 7.57 | −1.45 | 9.68 | −0.32 | 1.17 |
| Dairy products | 5.94 | −1.03 | 9.09 | 0.29 | −0.94 |
| Processed Rice | 4.36 | −0.85 | 5.62 | 0.26 | −0.31 |
| Other Food | 2.76 | −1.06 | 5.05 | 0.37 | −0.82 |
| Beverages & Tobacco | 4.11 | −1.52 | 4.23 | 0.53 | −0.60 |
| Petroleum & Coke | 25.17 | 24.51 | −59.54 | 0.46 | −0.38 |
| Chemicals | 27.75 | 22.62 | −71.70 | 1.88 | −3.85 |
| Motor vehicles and parts | 25.86 | 23.05 | −67.32 | 1.27 | −2.70 |
| Other Transport Equipment | 28.43 | 22.25 | −81.10 | 1.13 | −3.51 |
| Electronic Equipment | 25.25 | 23.10 | −75.08 | 6.80 | −13.00 |
| Other Machinery | 26.47 | 22.76 | −77.37 | 3.29 | −8.36 |
| Other Manufacturing | 26.52 | 22.53 | −70.15 | 5.96 | −11.28 |
| Services | 0.00 | −1.97 | 8.59 | 0.29 | −0.54 |

Source: Author's calculation.

Table 5 depicts the same variables, but now for China's economy. The first column in Scenario 2 shows the new tariffs for US products that have suffered Chinese retaliation. As expected, the price of US products facing higher import tariffs would rise in China, causing a significant drop in US exports to China and a rise in the price of Chinese aggregate imports in those sectors. Soybeans, for example, face an increase in import tariff from 2.42% to 27.42%. As a consequence, the price of soybeans would increase by 21.69% in this scenario. Given this increase in the price of US soybeans, its exports to China would decline 47.43%, leading to a fall of aggregate soybeans imports by China by 3.67%. Another sector retaliated by the Chinese that can be highlighted is dairy products. China raised the tariff by 25%, which resulted in a price increase of 23.6% and led to a reduction in China's imports from the US of 78.04% and a fall of Chinese aggregate imports in this sector by 9.44%.

**Table 5.** Chinese trade data (%).

| Sectors | China's Import Tariffs on US Exports (%) | China Import Prices of Product *i* from US | China Imports from US | China Aggregate Prices of Imports | China Aggregate Imports |
|---|---|---|---|---|---|
| | | Scenario 1 | | | |
| Iron & Steel | 2.29 | 0.83 | −7.20 | 0.25 | −3.95 |
| Aluminum | 0.95 | 1.03 | −11.42 | 0.28 | −5.69 |
| Soybeans | 2.42 | 0.17 | 0.21 | 0.25 | −0.17 |
| Primary products | 6.94 | 0.16 | −3.41 | 0.23 | −3.80 |
| Other not industrialized | 0.25 | 0.17 | −1.71 | 0.10 | −1.00 |
| Other industrialized | 4.26 | 0.28 | −5.45 | 0.23 | −5.12 |
| Dairy products | 6.29 | 0.16 | −4.62 | 0.19 | −4.81 |
| Processed Rice | 1.00 | 0.26 | −4.47 | 0.21 | −4.21 |
| Other Food | 10.82 | 0.15 | −3.10 | 0.21 | −3.34 |
| Beverages & Tobacco | 6.06 | 0.16 | −2.35 | 0.20 | −2.44 |
| Petroleum & Coke | 3.87 | 0.17 | −1.23 | 0.11 | −0.99 |
| Chemicals | 6.05 | 0.28 | −5.27 | 0.20 | −4.77 |
| Motor vehicles and parts | 22.43 | 0.54 | −6.28 | 0.25 | −4.77 |
| Other Transport Equipment | 2.55 | 0.45 | −8.72 | 0.29 | −7.45 |
| Electronic Equipment | 0.72 | 1.41 | −19.26 | 0.20 | −10.25 |
| Other Machinery | 4.83 | 0.53 | −8.82 | 0.22 | −6.47 |
| Other Manufacturing | 14.48 | 0.63 | −10.06 | 0.25 | −7.44 |
| Services | 0.00 | 0.13 | −3.99 | 0.28 | −4.53 |
| | | Scenario 2 | | | |
| Iron & Steel | 2.29 | 0.73 | −6.56 | 0.27 | −4.00 |
| Aluminum | 0.95 | 0.94 | −10.64 | 0.30 | −5.74 |
| Soybeans | 27.42 | 21.69 | −47.43 | 7.54 | −3.67 |
| Primary products | 31.94 | 22.86 | −67.78 | 3.13 | −9.74 |
| Other not industrialized | 0.25 | 0.12 | −0.80 | 0.15 | −1.03 |
| Other industrialized | 4.26 | 0.18 | −4.55 | 0.26 | −5.06 |
| Dairy products | 31.29 | 23.60 | −78.04 | 1.79 | −9.44 |
| Processed Rice | 26.00 | 24.91 | −69.01 | 0.27 | −2.86 |
| Other Food | 35.82 | 22.51 | −53.73 | 3.09 | −7.71 |
| Beverages & Tobacco | 31.06 | 23.61 | −37.59 | 3.18 | −5.44 |
| Petroleum & Coke | 3.87 | 0.16 | −1.06 | 0.14 | −1.02 |
| Chemicals | 6.05 | 0.18 | −4.50 | 0.21 | −4.69 |
| Motor vehicles and parts | 47.43 | 20.95 | −65.20 | 1.55 | −7.37 |
| Other Transport Equipment | 2.55 | 0.35 | −8.04 | 0.28 | −7.48 |
| Electronic Equipment | 0.72 | 1.32 | −18.53 | 0.22 | −10.30 |
| Other Machinery | 4.83 | 0.43 | −7.98 | 0.23 | −6.54 |
| Other Manufacturing | 14.48 | 0.53 | −9.27 | 0.26 | −7.45 |
| Services | 0.00 | 0.00 | −3.52 | 0.30 | −4.61 |

Source: Author's calculation.

The imposition of tariffs favored the US and Chinese trade balance, as shown in Table 6. In the first scenario, the US trade balance would increase $48.401 billion and $7.624 billion in China. Allowing for Chinese retaliation would not change much the results for these countries, as the US accumulate an increase of its trade balance of $52.154 billion and China of $10.678 billion. Thus, Scenario 1 would already be responsible for a large impact on the balance of trade and the US would achieve the objective of reducing its trade deficit through its tariff measures, and this result would be maintained even considering Chinese retaliation.

For the other regions, the result would be a reduction of their trade balances in both scenarios. The EU would be the most affected region, with its deficit increasing by $19.280 billion in scenario 1 and $21.302 billion in scenario 2. Emerging countries also would face a worsening of their trade balances, especially Brazil, with its deficit rising up to $6.457 billion in scenario 2.

**Table 6.** Trade balance ($ billion).

| Scenario 1 | | Scenario 2 | |
|---|---|---|---|
| China | 7.624 | China | 10.678 |
| US | 48.402 | US | 52.155 |
| Brazil | −4.528 | Brazil | −6.457 |
| Argentina | −0.913 | Argentina | −1.120 |
| India | −1.954 | India | −2.092 |
| Canada | −2.979 | Canada | −3.268 |
| Russia | −1.656 | Russia | −1.864 |
| Mexico | −3.099 | Mexico | −3.231 |
| EU | −19.280 | EU | −21.302 |
| Other | −21.612 | Other | −23.497 |

Source: Author's calculation.

Table 7 shows the trade balance by sectors. The sectors that contributed most to improving US trade balance in both scenarios would be iron & steel, electronic equipment, and other machinery, which showed a sharp decrease in aggregate imports, specially from China, due protectionist measures, as reported in Table 2. Regarding China, the most penalized sectors would be those characterized as high technology, especially electronic equipment, which were those targeted by tariff increases in the US. On the other hand, there would be an increase in trade balance of sectors not particularly penalized by trade restrictive measures in both scenarios, such as other industrialized products, which would allow for the Chinese surplus. The other regions, including emerging countries, in contrast, would face a reduction in trade balances in most sectors. The exception would be in some sectors the US and China increase tariffs against each other, but exempting others, like iron & steel and aluminum in both scenarios, and soybeans in scenario 2.

**Table 7.** Trade balance by sector ($ billion).

| Sectors | China | US | Brazil | Argentina | India | Canada | Russia | Mexico | EU | Other |
|---|---|---|---|---|---|---|---|---|---|---|
| **Scenario 1** | | | | | | | | | | |
| Iron & Steel | 2.388 | 7.820 | 1.729 | 0.146 | −0.948 | 2.821 | −1.084 | 0.771 | −4.647 | −8.473 |
| Aluminum | 6.022 | 1.546 | 0.236 | 0.205 | −0.108 | 3.544 | −0.942 | 1.125 | −2.569 | −8.989 |
| Soybeans | 0.015 | 0.047 | −0.032 | 0.006 | 0.002 | −0.080 | 0.000 | 0.043 | −0.011 | 0.024 |
| Primary products | 2.945 | 0.262 | −0.456 | −0.164 | −0.139 | −0.639 | 0.043 | −0.592 | −0.175 | −0.998 |
| Other not industrialized | 4.104 | −1.446 | −1.049 | −0.074 | −0.442 | −1.615 | 0.419 | −1.414 | −0.323 | 2.368 |
| Other industrialized | 47.152 | −5.526 | −1.808 | −0.229 | −1.552 | −3.928 | −0.423 | −3.916 | −14.521 | −16.894 |
| Dairy products | 0.159 | 0.080 | −0.008 | −0.013 | −0.002 | −0.022 | 0.007 | −0.137 | 0.028 | −0.096 |
| Processed Rice | 0.036 | 0.000 | −0.008 | 0.000 | 0.005 | 0.001 | 0.000 | −0.003 | −0.001 | −0.032 |
| Other Food | 2.020 | 0.149 | −0.074 | −0.043 | −0.017 | −0.369 | −0.035 | −0.535 | −0.342 | −0.841 |
| Beverages & Tobacco | 0.120 | 0.039 | −0.014 | 0.000 | 0.000 | −0.069 | 0.001 | −0.100 | 0.085 | −0.065 |
| Petroleum & Coke | 0.307 | 0.324 | −0.077 | −0.009 | 0.052 | −0.156 | 0.280 | −0.494 | −0.039 | −0.227 |
| Chemicals | 0.290 | 4.648 | −0.563 | −0.146 | 0.090 | 0.162 | 0.199 | −2.122 | 3.206 | −4.556 |
| Motor vehicles and parts | 0.475 | 2.702 | −0.122 | −0.131 | 0.000 | −0.389 | −0.078 | −3.153 | 0.600 | 0.415 |
| Other Transport Equipment | 4.161 | −1.290 | −0.191 | −0.018 | −0.057 | −0.558 | −0.002 | −0.428 | −0.415 | −1.181 |
| Electronic Equipment | −48.810 | 17.819 | −0.452 | −0.045 | −0.031 | 1.223 | −0.069 | 10.430 | −1.308 | 21.695 |
| Other Machinery | −10.158 | 13.149 | −0.848 | −0.164 | −0.257 | 0.601 | −0.234 | −0.399 | 1.993 | −2.645 |
| Other Manufacturing | −15.924 | 6.072 | −0.014 | −0.018 | 2.098 | 0.136 | 0.079 | −0.007 | 2.163 | 6.105 |
| Services | 12.322 | 2.017 | −0.778 | −0.217 | −0.650 | −3.641 | 0.182 | −2.167 | −3.005 | −7.226 |
| Total | 7.624 | 48.412 | −4.528 | −0.913 | −1.955 | −2.979 | −1.656 | −3.099 | −19.281 | −21.616 |
| **Scenario 2** | | | | | | | | | | |
| Iron & Steel | 2.437 | 8.022 | 1.560 | 0.129 | −0.952 | 2.830 | −1.101 | 0.780 | −4.661 | −8.520 |
| Aluminum | 6.056 | 1.945 | 0.076 | 0.147 | −0.117 | 3.562 | −0.977 | 1.155 | −2.652 | −9.126 |
| Soybeans | 0.570 | −5.388 | 3.241 | 1.225 | 0.001 | 0.100 | 0.000 | 0.096 | −0.028 | 0.361 |
| Primary products | 4.575 | −1.799 | −1.194 | −0.856 | 0.117 | −0.637 | 0.038 | −0.632 | 0.245 | 0.438 |
| Other not industrialized | 4.072 | −0.672 | −1.987 | −0.110 | −0.531 | −1.601 | 0.528 | −1.380 | −0.486 | 2.739 |
| Other industrialized | 46.028 | −3.480 | −2.464 | −0.310 | −1.566 | −4.000 | −0.449 | −3.929 | −14.353 | −17.035 |
| Dairy products | 0.273 | −0.136 | −0.015 | −0.025 | −0.002 | −0.022 | 0.005 | −0.140 | 0.057 | 0.012 |
| Processed Rice | 0.024 | 0.013 | −0.016 | −0.003 | 0.007 | 0.001 | 0.000 | −0.003 | −0.001 | −0.027 |

**Table 7.** *Cont.*

| Sectors | China | US | Brazil | Argentina | India | Canada | Russia | Mexico | EU | Other |
|---|---|---|---|---|---|---|---|---|---|---|
| | | | | Scenario 2 | | | | | | |
| Other Food | 2.050 | −0.452 | −0.134 | −0.070 | −0.012 | −0.366 | 0.042 | −0.527 | −0.187 | −0.385 |
| Beverages & Tobacco | 0.221 | −0.099 | −0.030 | −0.004 | 0.000 | −0.071 | 0.000 | −0.102 | 0.142 | −0.052 |
| Petroleum & Coke | 0.314 | 0.534 | −0.121 | −0.023 | 0.053 | −0.156 | 0.277 | −0.494 | −0.111 | −0.324 |
| Chemicals | −0.148 | 6.986 | −1.110 | −0.272 | 0.047 | 0.101 | 0.158 | −2.109 | 2.603 | −5.063 |
| Motor vehicles and parts | 1.798 | −0.227 | −0.320 | −0.214 | 0.002 | −0.512 | −0.103 | −3.208 | 2.013 | 1.122 |
| Other Transport Equipment | 4.202 | −0.415 | −0.327 | −0.030 | −0.070 | −0.574 | −0.021 | −0.431 | −0.790 | −1.520 |
| Electronic Equipment | −48.517 | 19.123 | −0.651 | −0.065 | −0.050 | 1.202 | −0.084 | 10.387 | −1.603 | 20.714 |
| Other Machinery | −9.726 | 16.800 | −1.568 | −0.243 | −0.316 | 0.502 | −0.329 | −0.503 | 0.700 | −4.276 |
| Other Manufacturing | −15.945 | 6.496 | −0.047 | −0.024 | 2.053 | 0.127 | 0.070 | −0.009 | 2.047 | 5.924 |
| Services | 12.395 | 4.912 | −1.349 | −0.371 | −0.756 | −3.750 | 0.081 | −2.183 | −4.241 | −8.483 |
| Total | 10.678 | 52.162 | −6.457 | −1.120 | −2.092 | −3.268 | −1.864 | −3.233 | −21.306 | −23.501 |

Source: Author's calculation.

Therefore, the results indicate that the new US trade strategy would be successful in reducing trade deficit, which has proven to be possible even in the scenario of Chinese retaliation. It would also guarantee better market access for US companies, as the trade balance would be positive mainly in the high technology sectors where most intellectual property rights are concentrated (another point justifying the US list of imposition Chinese products as a way of repairing the damage caused by misappropriation of US patents).

*3.3. Welfare*

In general equilibrium models based on perfect competition, with fixed endowments and technology, the only way to increase welfare is by reducing the excess burden caused by existing distortions with changes in allocative efficiency resulting from the interaction between tax and quantity changes.[12] Efficiency gains are closely related to the extent to which a country reduces its tariffs. Cheaper imported products cause gains in both consumption and in the way the domestic resources are employed. However, one may expect the contrary when countries are involved in a trade war, characterized by tariff increases. Changes in welfare are not restricted to allocative changes, but also include changes in terms of trade and the relative price of saving and investment (Hertel 1997).[13]

Figure 1 shows welfare measured in millions of dollars for the regions examined. The countries directly involved in the trade war would lose welfare in both simulations, with the second scenario accounting for the largest loss, reaching $23.598 billion in the US and $43.063 billion in China. Meanwhile, all remaining regions would benefit in both scenarios in terms of welfare, including emerging countries. Their gains are even larger in scenario 2, which considers Chinese retaliation.

Table 8 provides a decomposition of allocative efficiency effects for each region examined. Not surprisingly, the results confirm that the US, which increased its tariffs in scenario 1, is the country that suffered most, being responsible for the bulk of allocative losses when all regions are considered. Meanwhile, China welfare losses would be related mainly to terms of trade effect ($34.326 billion).

The terms of trade component dominate aggregate welfare changes for all regions but US in both scenarios. The improvement in US terms of trade component, partially offsetting allocative losses

---

[12]　The regional household's EV reflects the difference between the expenditure required to obtain the new level of utility at initial prices (YEV) and that level of utility available at the initial equilibrium (Y), that is to say EV = YEV−Y. According to Burfisher (2011), EV is a money metric measure which compares the cost of pre and post-shock levels of consumer utility, valued at base year prices. As a CGE model has a utility function, it is simple to calculate the utility obtained from different baskets of goods. For example, a tariff reduction in some good would cause a price change that allows the consumer to afford a new basket of goods that increases their utility. The EV measures the change in expenditure that consumers would have to get to afford the new level of utility at pre-shock prices.

[13]　The terms of trade (TOT) are defined as the ratio of the price received for tradeables to the price paid for them. McDougall (1993) shows that the change in terms of trade can be decomposed into three terms representing the contribution of world price indexes of all sectors, regional export and import prices. The impact on welfare, derived from the investment-savings component (I-S) depends on price of savings and investment and whether the region is either a net supplier or a net receiver of savings. The regions that are net suppliers of savings to the global bank benefit from an increase in price of savings relative to investment goods, while net receivers lose.

in scenario 1, results mainly from a fall in China's export prices to US, as the large tariff increases promoted by US reduced the demand for imports from China. This, in turn, deteriorates China's terms of trade. The other regions, including the emerging countries, would benefit from the higher US demand for their products bidding up their export prices, leading those regions to experience an improvement in their terms of trade. As a result, the aggregate welfare effect becomes positive in these countries in both simulations. The same logic applies for the second scenario, but as China retaliates the US, the latter country would now face a small terms of trade deterioration, as the demand for some of its products would decrease in China, leading US export prices falling.

Although the regions not directly involved may benefit from the trade war, the world's overall loss would be $25.446 billion in scenario 1 and $28.633 billion in scenario 2. As demonstrated above, this result was provoked mainly by the large allocative efficiency losses in US and terms of trade losses in China. This result is consistent with the recent literature dealing with economic effects of bilateral trade conflicts, where the use of tariffs leads to a reduction in the general welfare, being especially negative for the countries directly involved (Bollen and Rojas-Romagosa 2018; Tyner et al. 2018).

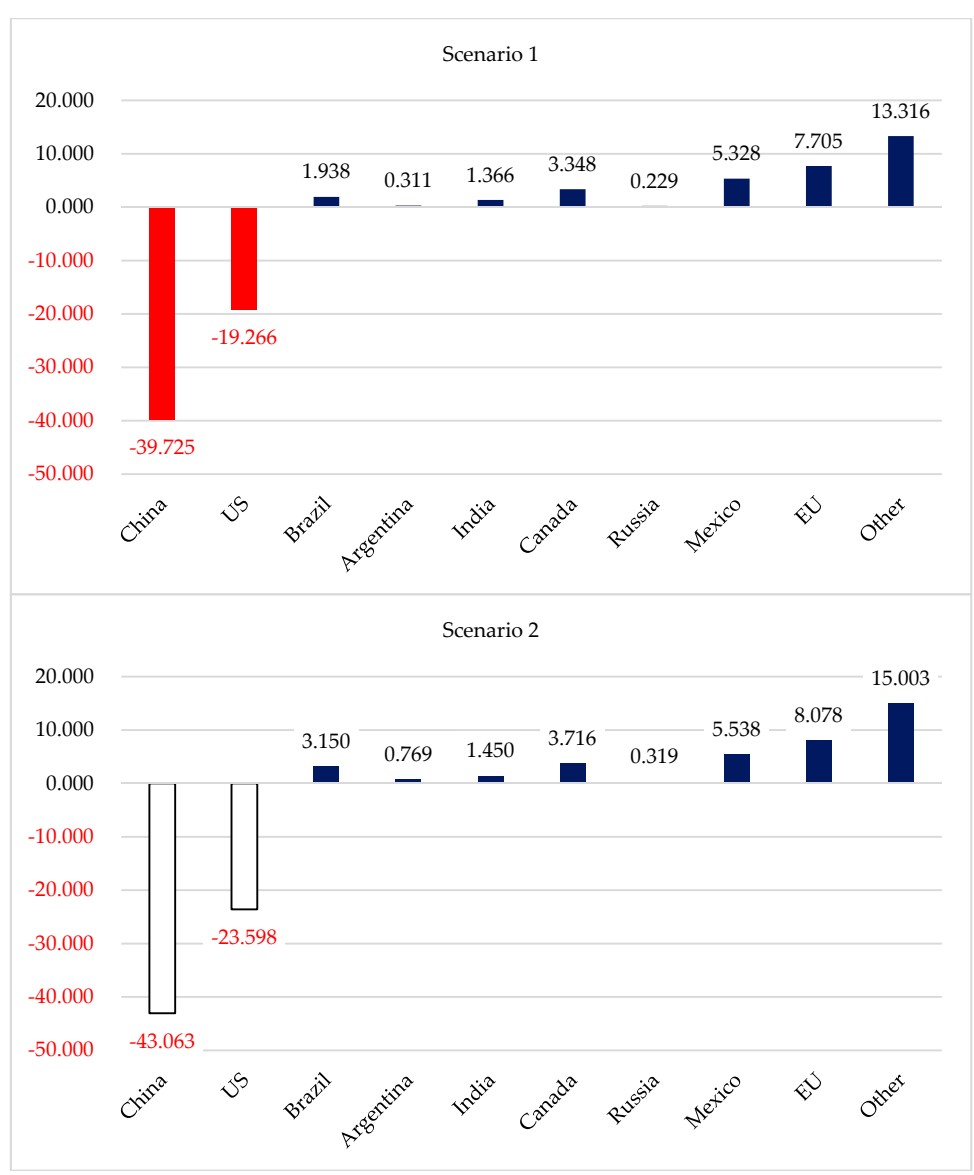

**Figure 1.** Effects of simulations on the welfare aggregate ($ billion). Source: Author's calculation.

**Table 8.** Effects on welfare ($ billion).

| Regions | Allocative Effects | Terms of Trade Effect | I-S Effect | Total |
|---|---|---|---|---|
| | | **Scenario 1** | | |
| China | −7.778 | −34.326 | 2.380 | −39.725 |
| US | −26.580 | 3.052 | 4.265 | −19.263 |
| Brazil | 1.025 | 1.071 | −0.158 | 1.938 |
| Argentina | 0.130 | 0.285 | −0.103 | 0.311 |
| India | 0.439 | 1.082 | −0.155 | 1.366 |
| Canada | 0.886 | 2.734 | −0.272 | 3.348 |
| Russia | 0.178 | 0.369 | −0.318 | 0.229 |
| Mexico | 0.276 | 5.727 | −0.675 | 5.328 |
| EU | 2.282 | 6.400 | −0.977 | 7.705 |
| Other | 4.070 | 13.202 | −3.956 | 13.316 |
| Total | −25.072 | −0.404 | 0.030 | −25.446 |
| | | **Scenario 2** | | |
| China | −11.812 | −33.975 | 2.724 | −43.063 |
| US | −26.423 | −0.504 | 3.332 | −23.595 |
| Brazil | 1.209 | 2.089 | −0.147 | 3.150 |
| Argentina | 0.226 | 0.673 | −0.130 | 0.769 |
| India | 0.435 | 1.118 | −0.103 | 1.450 |
| Canada | 0.928 | 3.029 | −0.242 | 3.716 |
| Russia | 0.195 | 0.433 | −0.308 | 0.319 |
| Mexico | 0.276 | 5.907 | −0.645 | 5.538 |
| EU | 2.197 | 6.719 | −0.837 | 8.078 |
| Other | 4.505 | 14.105 | −3.607 | 15.003 |
| Total | −28.263 | −0.406 | 0.036 | −28.634 |

Source: Author's calculation.

Table 9 provides a decomposition of allocative efficiency effects by commodity group for each region. It is possible to note that US experiences welfare losses over all commodities targeted by higher import tariffs on Chinese exports. The efficiency losses are concentrated exactly in the sectors more taxed, which are characterized by greater technological intensity, such as electronic equipment, other machinery, and motor vehicles and parts. These are the same sectors where there would be an increase in US domestic production and that would allow a surplus in the US trade balance, according to Tables 3 and 7. Therefore, the rise in US production would be associated with increased protectionism caused by the measures imposed by the US government.

With regard to China, the allocative losses are also concentrated in the high-tech sectors, where there would be a reduction in Chinese production, and an increase in trade deficits. The same would occur in primary sectors, in which China retaliated to the US, causing an increment in production, but inefficiently. Therefore, protectionism has led to large losses of allocative efficiency in both countries.

Table 10 shows the effect on the terms of trade by commodity group for each region. For both the US and China, the terms of trade effects would be dominated by high-technological sectors. In scenario 1, these sectors suffered a large loss in China due to lower demand in US, as a result of the unilateral tariff increases. In scenario 2, given the Chinese retaliation, those sectors also suffer a terms of trade deterioration in the US.

The results obtained in this section are similar to those of Tyner et al. (2018) and Ciuriak and Xiao (2018), which also showed welfare losses in the countries directly involved in the US–China trade war. The main contribution of this paper is to show the effects of those protectionist measures in developing countries, such as Argentina, Brazil, India, and Mexico, covering sectors of special interest to this group of countries, such as soybean and milk. With the increase in protectionism between the two largest global economies, it was possible to identify the sectors in which emerging countries, not directly involved in the trade war, could benefit by the shift in demand. Therefore, even if the trade dispute

generates losses, in terms of welfare and trade, for the US and China and for the world as a whole, certain sectors of emerging countries may benefit.

**Table 9.** Decomposition of allocative efficiency by sector ($ billion).

| Sectors | China | US | Brazil | Argentina | India | Canada | Russia | Mexico | EU | Other | Total |
|---|---|---|---|---|---|---|---|---|---|---|---|
| | | | | | **Scenario 1** | | | | | | |
| Iron & Steel | 0.018 | −1.583 | 0.098 | −0.029 | −0.095 | 0.032 | −0.055 | 0.009 | −0.118 | −0.173 | −1.897 |
| Aluminum | 0.040 | −0.673 | 0.013 | −0.038 | −0.040 | 0.046 | −0.055 | 0.021 | −0.062 | −0.077 | −0.824 |
| Soybeans | −0.003 | 0.000 | 0.001 | −0.004 | 0.000 | 0.000 | 0.000 | 0.000 | 0.000 | 0.049 | 0.043 |
| Primary products | −0.258 | −0.004 | 0.012 | −0.003 | 0.014 | 0.007 | −0.013 | 0.022 | 0.007 | 0.047 | −0.169 |
| Other not industrialized | 0.050 | 0.003 | −0.016 | −0.007 | −0.009 | −0.004 | 0.049 | −0.688 | −0.004 | −0.066 | −0.692 |
| Other industrialized | 3.501 | 0.718 | 0.110 | 0.029 | 0.020 | 0.143 | 0.076 | 0.175 | 0.491 | 1.471 | 6.734 |
| Dairy products | −0.038 | −0.001 | 0.004 | 0.002 | 0.002 | 0.023 | 0.000 | 0.015 | 0.016 | 0.009 | 0.033 |
| Processed Rice | −0.004 | 0.000 | 0.000 | 0.000 | −0.003 | 0.000 | 0.000 | 0.000 | 0.000 | 0.006 | 0.000 |
| Other Food | −0.129 | −0.002 | 0.010 | 0.001 | 0.003 | 0.020 | 0.003 | 0.029 | 0.040 | 0.138 | 0.114 |
| Beverages & Tobacco | −0.117 | −0.024 | 0.010 | 0.004 | 0.005 | 0.069 | 0.000 | 0.033 | 0.047 | 0.078 | 0.105 |
| Petroleum & Coke | −0.119 | −0.007 | 0.037 | 0.002 | 0.000 | 0.086 | 0.044 | 0.050 | 0.359 | −0.106 | 0.345 |
| Chemicals | −0.911 | −2.740 | 0.056 | 0.002 | 0.063 | 0.032 | 0.015 | 0.055 | 0.285 | 0.145 | −2.998 |
| Motor vehicles and parts | −1.319 | −0.842 | 0.134 | 0.033 | 0.041 | 0.067 | −0.007 | 0.083 | 0.163 | 0.326 | −1.321 |
| Other Transport Equipment | −0.173 | −0.388 | 0.015 | 0.011 | 0.038 | 0.008 | 0.008 | 0.021 | 0.039 | 0.255 | −0.165 |
| Electronic Equipment | −1.866 | −10.480 | 0.116 | 0.016 | 0.030 | 0.030 | 0.005 | 0.119 | 0.088 | 0.411 | −11.531 |
| Other Machinery | −3.544 | −6.858 | 0.237 | 0.036 | 0.221 | 0.046 | 0.065 | 0.129 | 0.427 | 0.484 | −8.756 |
| Other Manufacturing | −0.654 | −3.047 | 0.040 | 0.008 | 0.134 | 0.030 | 0.014 | 0.061 | 0.239 | 0.192 | −2.983 |
| Services | −2.070 | −0.663 | 0.136 | 0.067 | 0.013 | 0.221 | 0.031 | 0.085 | 0.277 | 0.768 | −1.134 |
| Total | −7.778 | −26.593 | 1.025 | 0.130 | 0.439 | 0.886 | 0.178 | 0.276 | 2.283 | 4.070 | −25.085 |
| | | | | | **Scenario 2** | | | | | | |
| Iron & Steel | 0.018 | −1.577 | 0.088 | −0.021 | −0.095 | 0.032 | −0.056 | 0.008 | −0.118 | −0.176 | −1.897 |
| Aluminum | 0.040 | −0.664 | 0.006 | −0.022 | −0.042 | 0.046 | −0.057 | 0.021 | −0.062 | −0.081 | −0.814 |
| Soybeans | −0.816 | 0.013 | −0.014 | 0.046 | 0.000 | 0.000 | 0.000 | 0.000 | 0.000 | 0.092 | −0.679 |
| Primary products | −1.158 | −0.010 | 0.014 | −0.037 | 0.015 | 0.019 | −0.008 | 0.019 | −0.017 | 0.439 | −0.725 |
| Other not industrialized | 0.041 | 0.041 | −0.036 | −0.019 | −0.010 | −0.005 | 0.052 | −0.675 | −0.005 | −0.069 | −0.684 |
| Other industrialized | 3.339 | 0.615 | 0.123 | 0.032 | 0.017 | 0.137 | 0.072 | 0.168 | 0.473 | 1.439 | 6.417 |
| Dairy products | −0.091 | −0.005 | 0.006 | 0.003 | 0.002 | 0.025 | 0.000 | 0.014 | 0.015 | 0.013 | −0.017 |
| Processed Rice | −0.004 | 0.000 | 0.000 | −0.001 | −0.003 | 0.000 | 0.000 | 0.000 | 0.000 | 0.013 | 0.005 |
| Other Food | −0.390 | −0.013 | 0.012 | 0.001 | 0.003 | 0.024 | 0.003 | 0.028 | 0.034 | 0.123 | −0.175 |
| Beverages & Tobacco | −0.169 | −0.034 | 0.016 | 0.007 | 0.005 | 0.074 | 0.000 | 0.034 | 0.051 | 0.082 | 0.067 |
| Petroleum & Coke | −0.143 | −0.042 | 0.054 | 0.009 | −0.004 | 0.088 | 0.040 | 0.051 | 0.318 | −0.117 | 0.252 |
| Chemicals | −0.919 | −2.710 | 0.067 | 0.004 | 0.059 | 0.032 | 0.015 | 0.054 | 0.263 | 0.141 | −2.994 |
| Motor vehicles and parts | −2.484 | −0.899 | 0.176 | 0.044 | 0.043 | 0.069 | −0.002 | 0.080 | 0.267 | 0.363 | −2.343 |
| Other Transport Equipment | −0.182 | −0.383 | 0.015 | 0.013 | 0.040 | 0.008 | 0.009 | 0.020 | 0.036 | 0.251 | −0.173 |
| Electronic Equipment | −1.885 | −10.478 | 0.157 | 0.019 | 0.030 | 0.031 | 0.006 | 0.119 | 0.073 | 0.404 | −11.524 |
| Other Machinery | −3.632 | −6.842 | 0.308 | 0.044 | 0.227 | 0.047 | 0.071 | 0.125 | 0.407 | 0.490 | −8.755 |
| Other Manufacturing | −0.661 | −3.061 | 0.052 | 0.010 | 0.133 | 0.030 | 0.014 | 0.061 | 0.235 | 0.193 | −2.993 |
| Services | −2.310 | −0.639 | 0.178 | 0.095 | 0.014 | 0.243 | 0.038 | 0.089 | 0.286 | 0.830 | −1.177 |
| Total | −11.812 | −26.437 | 1.209 | 0.226 | 0.435 | 0.928 | 0.195 | 0.276 | 2.197 | 4.505 | −28.276 |

Source: Author's calculation.

**Table 10.** Variation in terms of trade by sector ($ billion).

| Sectors | China | US | Brazil | Argentina | India | Canada | Russia | Mexico | EU | Other | Total |
|---|---|---|---|---|---|---|---|---|---|---|---|
| | | | | | **Scenario 1** | | | | | | |
| Iron & Steel | −0.723 | 0.095 | 0.053 | 0.002 | 0.034 | 0.032 | 0.006 | 0.081 | 0.089 | 0.336 | 0.004 |
| Aluminum | −0.514 | 0.171 | 0.022 | 0.013 | −0.028 | 0.198 | −0.001 | 0.284 | −0.192 | 0.077 | 0.029 |
| Soybeans | −0.092 | 0.019 | 0.051 | 0.012 | 0.002 | 0.033 | −0.001 | −0.007 | −0.014 | −0.021 | −0.018 |
| Primary products | −0.387 | −0.043 | 0.130 | 0.078 | 0.014 | 0.138 | −0.021 | 0.120 | −0.048 | 0.067 | 0.047 |
| Other not industrialized | −0.210 | −0.154 | 0.094 | 0.005 | −0.018 | 0.185 | 0.010 | 0.170 | −0.122 | 0.134 | 0.095 |
| Other industrialized | −9.229 | 0.931 | 0.243 | 0.034 | 0.218 | 0.461 | 0.204 | 0.478 | 2.003 | 4.095 | −0.563 |
| Dairy products | −0.007 | 0.001 | −0.001 | 0.006 | 0.000 | 0.002 | −0.003 | 0.000 | 0.011 | −0.009 | −0.001 |
| Processed Rice | −0.005 | 0.002 | 0.002 | 0.001 | 0.008 | 0.000 | 0.000 | 0.000 | −0.001 | −0.008 | −0.001 |
| Other Food | −0.590 | −0.041 | 0.022 | 0.020 | 0.014 | 0.097 | 0.006 | 0.129 | 0.071 | 0.313 | 0.042 |
| Beverages & Tobacco | −0.033 | −0.044 | 0.006 | 0.005 | 0.001 | 0.007 | −0.003 | 0.074 | 0.031 | −0.008 | 0.036 |
| Petroleum & Coke | −0.210 | 0.036 | −0.001 | 0.000 | 0.036 | 0.037 | 0.004 | 0.042 | −0.031 | 0.106 | 0.019 |
| Chemicals | −2.936 | 0.264 | 0.050 | 0.022 | 0.162 | 0.323 | 0.013 | 0.286 | 0.526 | 1.251 | −0.039 |
| Motor vehicles and parts | −0.786 | −0.324 | −0.002 | −0.028 | 0.017 | 0.116 | −0.021 | 1.138 | 0.121 | 0.194 | 0.425 |
| Other Transport Equipment | −0.796 | 0.204 | 0.012 | 0.006 | 0.022 | 0.098 | −0.002 | 0.062 | 0.072 | 0.299 | −0.023 |
| Electronic Equipment | −6.314 | 1.117 | 0.092 | 0.021 | 0.133 | 0.058 | 0.103 | 1.182 | 1.214 | 1.982 | −0.412 |
| Other Machinery | −6.239 | 0.736 | 0.161 | 0.026 | 0.171 | 0.126 | 0.086 | 1.179 | 1.248 | 2.434 | −0.072 |
| Other Manufacturing | −1.590 | 0.271 | 0.020 | 0.009 | 0.079 | 0.051 | 0.032 | 0.060 | 0.497 | 0.425 | −0.147 |
| Services | −3.665 | −0.188 | 0.117 | 0.053 | 0.216 | 0.774 | −0.044 | 0.451 | 0.925 | 1.536 | 0.175 |
| Total | −34.328 | 3.053 | 1.071 | 0.285 | 1.083 | 2.734 | 0.369 | 5.729 | 6.400 | 13.202 | −0.403 |
| | | | | | **Scenario 2** | | | | | | |
| Iron & Steel | −0.730 | −0.103 | 0.085 | 0.003 | 0.034 | 0.038 | 0.009 | 0.084 | 0.089 | 0.343 | −0.148 |
| Aluminum | −0.517 | −0.218 | 0.041 | 0.020 | −0.030 | 0.196 | 0.002 | 0.284 | −0.182 | 0.099 | −0.305 |
| Soybeans | −0.102 | 0.740 | 0.335 | 0.118 | 0.004 | 0.055 | −0.003 | 0.048 | −0.018 | 0.097 | 1.274 |
| Primary products | −0.301 | 0.861 | 0.320 | 0.275 | 0.021 | 0.191 | −0.037 | 0.179 | −0.132 | 0.162 | 1.540 |
| Other not industrialized | −0.314 | 0.337 | 0.201 | 0.008 | −0.032 | 0.187 | 0.040 | 0.169 | −0.200 | 0.216 | 0.613 |

**Table 10.** *Cont.*

| Sectors | China | US | Brazil | Argentina | India | Canada | Russia | Mexico | EU | Other | Total |
|---|---|---|---|---|---|---|---|---|---|---|---|
| | | | | | Scenario 2 | | | | | | |
| Other industrialized | −8.992 | −1.331 | 0.340 | 0.035 | 0.220 | 0.483 | 0.209 | 0.491 | 1.957 | 4.006 | −2.582 |
| Dairy products | −0.007 | 0.006 | −0.002 | 0.009 | 0.000 | 0.002 | −0.003 | 0.002 | 0.012 | −0.009 | 0.010 |
| Processed Rice | −0.003 | 0.001 | 0.004 | 0.002 | 0.010 | 0.000 | 0.000 | 0.000 | −0.001 | −0.011 | 0.000 |
| Other Food | −0.437 | 0.192 | 0.037 | 0.036 | 0.018 | 0.108 | 0.002 | 0.129 | 0.048 | 0.238 | 0.371 |
| Beverages & Tobacco | −0.030 | 0.104 | 0.013 | 0.010 | 0.001 | 0.010 | −0.004 | 0.073 | 0.037 | −0.010 | 0.205 |
| Petroleum & Coke | −0.210 | 0.017 | 0.006 | 0.002 | 0.044 | 0.041 | 0.012 | 0.049 | −0.027 | 0.121 | 0.055 |
| Chemicals | −2.867 | 0.009 | 0.099 | 0.032 | 0.164 | 0.356 | 0.017 | 0.317 | 0.601 | 1.325 | 0.053 |
| Motor vehicles and parts | −0.745 | 0.740 | 0.023 | −0.033 | 0.017 | 0.153 | −0.024 | 1.134 | 0.136 | 0.206 | 1.608 |
| Other Transport Equipment | −0.797 | −0.197 | 0.027 | 0.006 | 0.023 | 0.104 | −0.001 | 0.062 | 0.113 | 0.353 | −0.307 |
| Electronic Equipment | −6.374 | −1.815 | 0.098 | 0.021 | 0.135 | 0.065 | 0.114 | 1.174 | 1.248 | 2.082 | −3.253 |
| Other Machinery | −6.275 | −0.736 | 0.212 | 0.027 | 0.175 | 0.178 | 0.100 | 1.198 | 1.351 | 2.632 | −1.139 |
| Other Manufacturing | −1.583 | −0.420 | 0.022 | 0.009 | 0.083 | 0.054 | 0.035 | 0.061 | 0.506 | 0.436 | −0.798 |
| Services | −3.693 | 1.311 | 0.227 | 0.092 | 0.231 | 0.810 | −0.034 | 0.456 | 1.181 | 1.819 | 2.400 |
| Total | −33.977 | −0.504 | 2.089 | 0.674 | 1.118 | 3.030 | 0.433 | 5.909 | 6.719 | 14.106 | −0.404 |

Source: Author's calculation.

## 3.4. Sensitivity Analysis

One of the most recurring criticisms of the general equilibrium models is the strong dependence of their results on the value of elasticities of substitution. According to Domingues et al. (2008), many applications employ parameters that require more precise estimates. One way to try to mitigate such criticisms by examining the robustness of the results is through tests that expose the sensitivity of the model to variations in the adopted parameters. To this end, the GTAP provides the Systematic Sensitivity Analysis tool.

This analysis consists of varying the values of the substitution elasticities within a range, with the model being solved innumerable times, generating averages, standard deviations, and confidence intervals for the results of interest (Wigle 1991). If there is a significant change in the confidence intervals in terms of their amplitude, it is a sign that the model would not be robust and may even generate changes in the signal of the variable examined.

The parameters that are usually subject to variations for the sensitivity test are the elasticity of substitution between domestic inputs (ESUBD), the elasticity of substitution between domestic and imported inputs (ESUBD), and the elasticity of substitution between primary production factors (ESUBVA) (Wigle 1991). In this study, for both scenarios, the parameters ESUBD, ESUBT, and ESUBVA were varied by ±50%. The endogenous variable chosen for the analysis was the EV welfare indicator (equivalent variation of regional consumer income), whose 93.75% confidence interval was determined using Chebyshev's inequality.

Table 11 shows the results for the two scenarios simulated in the study. In scenario 1, only Russia shows signal inversion between the lower and upper limits of the confidence interval, but the negative value found is very close to zero. All the other regions examined present the same signal between the lower and upper limits, indicating the robustness of the model. China and US welfare losses could reach up to $50.9 billion and $21.2 billion, respectively. In the case of Brazil, welfare gains could reach $2.7 billion, while the EU would potentially be the most benefited region, with gains of up to $10.1 billion. Total welfare losses could reach $49.6 billion in this scenario.

In Scenario 2, with Chinese retaliation, there would be no ambiguity in the results found, but global losses would increase, reaching up to $54.7 billion. Again, China would be the most impaired, with potential losses of $54 billion, with the US making losses that could reach $26.1 billion. The losses of the two countries directly involved in the trade war would bring benefits to the other regions, which remained outside the direct application of the protectionist measures. For all emerging countries examined, gains would be increased in comparison with Scenario 1, especially Argentina and Brazil.

**Table 11.** Sensitivity analysis on welfare ($ billion).

| Region | Scenario 1 | | | | Scenario 2 | | | |
|---|---|---|---|---|---|---|---|---|
| | Mean | Standard Deviation | 93.75 Confidence Interval | | Mean | Standard Deviation | 93.75 Confidence Interval | |
| China | −39.394 | 2.887 | −50.942 | −27.846 | −42.703 | 2.828 | −54.014 | −31.392 |
| US | −19.066 | 0.533 | −21.197 | −16.936 | −23.403 | 0.685 | −26.144 | −20.661 |
| Brazil | 1.957 | 0.183 | 1.227 | 2.687 | 3.177 | 0.297 | 1.988 | 4.365 |
| Argentina | 0.317 | 0.043 | 0.146 | 0.487 | 0.779 | 0.103 | 0.367 | 1.191 |
| India | 1.357 | 0.143 | 0.786 | 1.927 | 1.441 | 0.145 | 0.861 | 2.021 |
| Canada | 3.354 | 0.340 | 1.996 | 4.713 | 3.723 | 0.368 | 2.252 | 5.194 |
| Russia | 0.234 | 0.059 | −0.002 | 0.469 | 0.325 | 0.068 | 0.054 | 0.596 |
| Mexico | 5.288 | 0.304 | 4.071 | 6.505 | 5.499 | 0.318 | 4.228 | 6.769 |
| EU | 7.658 | 0.616 | 5.194 | 10.122 | 8.036 | 0.665 | 5.378 | 10.694 |
| Other Countries | 13.249 | 1.033 | 9.116 | 17.382 | 14.944 | 1.151 | 10.340 | 19.549 |
| Total | −25.046 | 6.140 | −49.606 | −0.486 | −28.182 | 6.627 | −54.690 | −1.674 |

Source: Author's calculation.

## 4. Discussion

Studies examining the impacts of trade war between the US and China using CGE models are still rare because these trade policies are very recent. Nevertheless, some exercises have already been made. Ciuriak and Xiao (2018), for example, quantified the impacts of Section 232 of the steel and aluminum tariffs, through the GTAP, for the US, China, and other developed countries, such as the EU, Canada, Japan, South Korea, and Mexico. The simulations did not involve retaliation and showed only the implications for US trade, global macroeconomic impacts, and sectoral impacts. The main conclusions would be the increase in the prices of these products in the USA, consequently a reduction in the exports of these products and derived products, due to the loss of competitiveness; decrease in real GDP by 0.06% and a drop on welfare of $6.3 billion. With respect to the other countries, Mexico and Canada would present losses, but the country that would suffer the greatest negative impact would be Canada, with GDP falling 0.11% and a loss of welfare of $3.7 billion. Other US trading partners, on the other hand, would not be globally affected, as damage to the competitiveness of US trade would boost gains for China, Japan, the EU, and South Korea.

Bollen and Rojas-Romagosa (2018) used WorldScan, a computable general equilibrium model of Netherlands Bureau for Economic Policy Analysis, whose database is the GTAP version 9, for the analysis of the economic consequences of the current conflict between the US and several countries that have announced retaliatory packages at the WTO. The authors allowed for a wide range of retaliation scenarios, following the US tariffs, from China, Canada, Mexico, the EU, and the rest of the OECD countries. The paper concludes that for the scenario with the unilateral imposition of tariffs on steel and aluminum, the impact on the US and its trading partners would be small, with the most negative effects occurring for industries that use these products as intermediate inputs, especially in the sectors of electronic equipment, other machinery and equipment, and agriculture. In retaliation scenarios by trading partners, the results vary widely by region and global effects would be limited, but they emphasize that the economic effects of bilateral trade conflicts would be negative for the countries involved. However, in the US–China trade dispute scenario, the result would be asymmetric, with China having a significant GDP loss of 1.2%, while the US would have a loss limited to 0.3%, justified by US market power and of the US trade deficit with China. In addition, they conclude that for the US, the number of countries involved in the conflict would be important, because the losses would be greater when the conflict extends beyond the EU and China, particularly when involving Canada and Mexico.

Tyner et al. (2018) presented a brief report on the quantitative evaluation of the possible impacts of the new agreement of the North American Free Trade Agreement (NAFTA), known as the US–Mexico–Canada Agreement (USMCA), in US agriculture. The authors use different contexts, including the policy of imposing 25% and 10% increases in steel and aluminum tariffs, respectively, with US trading partners reacting to these tariffs, targeting US exports in sensitive sectors such as

agriculture. According to their simulations using GTAP, retaliatory tariffs implemented by Canada and Mexico would reverse modest export earnings with the USMCA—a decline of $1.7 billion instead of a $450 million gain. In another scenario, with broader and more global retaliation, US would suffer a fall in exports by about $8 billion, a reduction in both welfare ($27.8 billion) and GDP (0.08%).

In this section, we present some recent studies that aimed to measure the results of a trade war among the world's major economies due to a tariff escalation. Despite the different scenarios presented in these studies, they had some similar results, such as increase in US prices, small reduction in US GDP, slightly more significant decline in China's GDP, and reduction in both countries' welfare, which are confirmed by our results. Regarding welfare, our results are similar to those of Tyner et al. (2018) and Ciuriak and Xiao (2018), showing welfare losses in the countries directly involved in the US–China trade war.[14]

However, our results point out that the new US trade policy would be successful in reducing its trade deficit, the main goal of President Trump, even in the scenario of Chinese retaliation. It would also guarantee better market access for US companies, as the trade balance would be mainly positive in the high technology sectors where most intellectual property rights are concentrated, which is another concern of the current US administration.

Finally, the main contribution of this paper is to show the effects of those protectionist measures for developing countries. With the increase in protectionism between US and China, some sectors in emerging countries, not directly involved in the trade war, could benefit by the shift in demand, despite the overall losses in terms of welfare for the US and China and for the world as a whole. As a result, all emerging countries examined experienced welfare gains in both scenarios, especially Mexico and Brazil.

## 5. Conclusions

The results show that the US–China trade war would have the desired effect of President Trump, because it would reduce the country's trade deficit, even with Chinese retaliation, by around $50 billion. In addition, in both scenarios, there would be an increase in the production of the steel and aluminum sectors in the United States, preferential targets of the protectionist measures of that country.

However, if the trade war would lead to a reduction in US imports of Chinese products on which it imposed tariffs, improving trade balance and domestic production, Chinese retaliation would reduce the import of all US products. As a result, both countries and the world as a whole would lose in terms of welfare, due to the significant reduction in allocative efficiency, especially in the US, and the loss of terms of trade in the Chinese case. However, Chinese producers and consumers would bear the lion's share of the burden of the trade war launched by the US. It is worth noting that the effects are more significant in the first scenario, that is, when the US imposes the additional import tariff. When China retaliates, the tariff effects would occur to a lesser extent. Therefore, the weight of the first country to impose the tariff, the US, would be much higher in a tariff war.

Regarding the impacts on emerging countries, there would a reduction in trade balances in most sectors, in contrast to the US and China. However, those countries benefit from a rise in welfare in both scenarios. Their gains are even larger in scenario 2, which considers Chinese retaliation. The welfare gains are concentrated in terms of trade improvements as they would benefit from higher US and Chinese demand for their products, bidding up their export prices to these countries. In the case of Argentina and Brazil, the gains are larger in primary sectors, including soybeans, while in Mexico and India, industrialized sectors benefit the most, such as electronic equipment and other machinery.

**Author Contributions:** Conceptualization, M.C. and A.A.; Data curation, M.C., A.A., and A.M.; Formal analysis, M.C., A.A., and A.M.; Funding acquisition, A.A.; Investigation, M.C., A.A., and A.M.; Methodology, M.C. and A.A.;

---

[14] Other studies based on CGE models, such as The International Monetary Fund (IMF 2018), Taheripour and Tyner (2018) and Li et al. (2018), also found that US and China would face losses due to the trade war, in terms of both GDP and welfare.

Project administration, A.A. and A.M.; Resources, M.C., A.A., and A.M.; Software, M.C. and A.A.; Supervision, A.A. and A.M.; Validation, M.C., A.A., and A.M.; Visualization, A.A. and A.M.; Writing—original draft, M.C.; Writing—review & editing, A.A. and A.M.

**Funding:** This research was funded by Conselho Nacional de Desenvolvimento Científico e Tecnológico (4078382018-0).

**Conflicts of Interest:** The authors declare no conflict of interest.

## Appendix A

**Table A1.** Elasticities of Substitution.

| GTAP Sector | ESUBD [1] | ESUBM [2] | ESUBVA [3] |
|---|---|---|---|
| Iron & Steel | 2.95 | 5.90 | 1.26 |
| Aluminum | 4.20 | 8.40 | 1.26 |
| Soybeans | 2.45 | 4.90 | 0.26 |
| Primary products | 2.78 | 5.88 | 0.38 |
| Other not industrialized | 4.85 | 11.41 | 0.21 |
| Other industrialized | 3.40 | 7.04 | 1.26 |
| Dairy products | 3.65 | 7.30 | 1.12 |
| Processed Rice | 2.60 | 5.20 | 1.12 |
| Other Food | 2.00 | 4.00 | 1.12 |
| Beverages & Tobacco | 1.15 | 2.30 | 1.12 |
| Petroleum & Coke | 2.10 | 4.20 | 1.26 |
| Chemicals | 3.30 | 6.60 | 1.26 |
| Motor vehicles and parts | 2.80 | 5.60 | 1.26 |
| Other Transport Equipment | 4.30 | 8.60 | 1.26 |
| Electronic Equipment | 4.40 | 8.80 | 1.26 |
| Other Machinery | 4.05 | 8.10 | 1.26 |
| Other Manufacturing | 3.75 | 7.50 | 1.26 |
| Services | 1.94 | 3.85 | 1.36 |

Source: Author's calculation. 1: values of the elasticity of substitution between domestic and imported goods in the Armington aggregation structure; 2: values of the elasticity of substitution between imports from different sources; 3: values of the elasticity of substitution between primary factors.

## Appendix B

**Table A2.** GTAP aggregation.

| Initials | Product | Sector Aggregation |
|---|---|---|
| i_s | Ferrous metals | Iron & Steel |
| nfm | Metals nec | Aluminum |
| osd | Oil seeds | Soybeans |
| pdr | Paddy rice | Primary products |
| wht | Wheat | |
| gro | Cereal grains nec | |
| v_f | Vegetables, fruit, nuts | |
| pfb | Plant-based fibers | |
| ocr | Crops nec | |
| oap | Animal products nec | |
| fsh | Fishing | |
| omt | Meat products nec | |
| vol | Vegetable oils and fats | |
| c_b | Sugar cane, sugar beet | Other not industrialized |
| ctl | Cattle, sheep, goats, horses | |
| rmk | Raw milk | |
| wol | Wool, silk-worm cocoons | |
| frs | Forestry | |
| coa | Coal | |

**Table A2.** *Cont.*

| Initials | Product | Sector Aggregation |
|---|---|---|
| oil | Oil | |
| gas | Gas | |
| omn | Minerals nec | |
| sgr | Sugar | Other industrialized |
| tex | Textiles | |
| wap | Wearing apparel | |
| lea | Leather products | |
| lum | Wood products | |
| ppp | Paper products, publishing | |
| nmm | Mineral products nec | |
| fmp | Metal products | |
| mil | Dairy products | Dairy products |
| pcr | Processed rice | Processed Rice |
| ofd | Food products nec | Other Food |
| b_t | Beverages and tobacco products | Beverages & Tobacco |
| p_c | Petroleum, coal products | Petroleum & Coke |
| crp | Chemical,rubber,plastic prods | Chemicals |
| mvh | Motor vehicles and parts | Motor vehicles and parts |
| otn | Transport equipment nec | Other Transport Equipment |
| ele | Electronic equipment | Electronic Equipment |
| ome | Machinery and equipment nec | Other Machinery |
| omf | Manufactures nec | Other Manufacturing |
| ely | Electricity | Services |
| gdt | Gas manufacture, distribution | |
| wtr | Water | |
| cns | Construction | |
| trd | Trade | |
| otp | Transport nec | |
| wtp | Sea transport | |
| atp | Air transport | |
| cmn | Communication | |
| ofi | Financial services nec | |
| isr | Insurance | |
| obs | Business services nec | |
| ros | Recreation and other services | |
| osg | PubAdmin/Defence/Health/Educat | |
| dwe | Dwellings | |

Source: Author's calculation.

## Appendix C

**Table A3.** GTAP codes & OCDE classification.

| N° | GTAP Sector | GTAP Code | GTA Description | OECD Classification | Tariff Increased by US | Tariff Increased by China |
|---|---|---|---|---|---|---|
| 1 | Iron & Steel | 35 | Iron & Steel: basic production and casting | medium-low-technology industries | Yes | No |
| 2 | Aluminum | 36 | Non-Ferrous Metals: production and casting of copper, aluminum, zinc, lead, gold, and silver | medium-low-technology industries | Yes | No |
| 3 | Soybeans | 5 | Oil Seeds: oil seeds and oleaginous fruit; soy beans, copra | Primary products | No | No |
| 4 | Primary products | 1/2/3/4/7/8/10/14/19/20/21 | Not Industrialized products which were target of import tariffs | Primary products | No | Yes |

**Table A3.** *Cont.*

| N° | GTAP Sector | GTAP Code | GTA Description | OECD Classification | Tariff Increased by US | Tariff Increased by China |
|---|---|---|---|---|---|---|
| 5 | Other not industrialized | 6/9/11/12/13/15/16/17/18 | Not Industrialized products which were not target of import tariffs | Primary products | No | No |
| 6 | Other industrialized | 24/27/28/29/30/31/34/37 | Industrialized products which were not target of import tariffs | low, medium-low and high technology industries | No | No |
| 7 | Dairy products | 22 | Milk: dairy products | low-technology industries | No | Yes |
| 8 | Processed Rice | 23 | Processed Rice: rice, semi- or wholly milled | low-technology industries | No | Yes |
| 9 | Other Food | 25 | Other Food: prepared and preserved fish or vegetables, fruit juices and vegetable juices, prepared and preserved fruit and nuts, all cereal flours, food products n.e.c. | low-technology industries | No | Yes |
| 10 | Beverages & Tobacco | 26 | Beverages and Tobacco products | low-technology industries | No | Yes |
| 11 | Petroleum & Coke | 32 | Petroleum & Coke: coke oven products, refined petroleum products, processing of nuclear fuel | medium-low-technology industries | Yes | No |
| 12 | Chemicals | 33 | Chemical Rubber Products: basic chemicals, other chemical products, rubber and plastics products | medium-high-technology industries | No | No |
| 13 | Motor vehicles and parts | 38 | Motor vehicles and parts: cars, lorries, trailers and semi-trailers | medium-high-technology industries | Yes | Yes |
| 14 | Other Transport Equipment | 39 | Other Transport Equipment: Manufacture of other transport equipment | medium-high-technology industries | Yes | No |
| 15 | Electronic Equipment | 40 | Electronic Equipment: office, accounting and computing machinery, radio, television and communication equipment and apparatus | high-technology industries | Yes | No |
| 16 | Other Machinery | 41 | Other Machinery & Equipment: electrical machinery and apparatus n.e.c., medical, precision and optical instruments, watches and clocks | high-technology industries | Yes | No |
| 17 | Other Manufacturing | 42 | Other Manufacturing: includes recycling | high-technology industries | Yes | No |
| 18 | Services | 43–57 | Services | Services | No | No |

Source: Author's calculation.

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
