# Peer review of "Emerging Countries and the Effects of the Trade War between US and China"

_economies, doi:10.3390/economies7020045_

Round 1

Reviewer 1 Report

The article is well-written, but I feel that the reference list is too short, 11 resources for the paper with 22 pages is not enought at all. I recommend adding some other resources, after that can be article published.

Author Response

We would like to thank the comments and suggestions of the reviewer, who have enriched the paper. They were all included in the new version of the paper and highlighted in red.

Point 1: The article is well-written, but I feel that the reference list is too short, 11 resources for the paper with 22 pages is not enought at all. I recommend adding some other resources, after that can be article published.

Response 1: New references dealing with the US-China trade war were included in footnote 14.

Reviewer 2 Report

The paper covers a very interesting and actual topic - the trade war between USA and China.

Despite the fact that the measures under analysis are quite recent, the authors manage to find impacts. The methodology is adequate and the conclusions obtained are interesting.

The structure of the paper is fine and the general importance of the topic discussed is high.

Author Response

We would like to thank the comments and suggestions of the reviewer, who have enriched the paper. They were all included in the new version of the paper and highlighted in red.

Reviewer 3 Report

I think that the methodology section should be improved to render the text more comprehensible for researchers less familiar with the GTAP CGE model, as well as the discussion of results section should be extended.

Firstly, more comments on the utility (especially EV, in the first occurrence of this abbreviation /p. 12, footnote 7/ it is not explained), three-level production function (mostly its relation to qo) and elasticities (used in the sensitivity analysis) should be provided. The term "savings prices" should be exemplified as well. Was the model computed with the help of RunGTAP? From the text it seems so but any non-standard GEMPACK computation must be documented.

If I may, I would also suggest using the terms "income and substitution effect" for the effects defined on pages 3 and 4, lines 128-139.

Secondly, I have a few equations regarding the equations on pages 3 and 4. Where were they taken from? The general description of GTAP variables does not mention any of them: https://www.gtap.agecon.purdue.edu/models/setsvariables.asp Was it the Manual or the TABLO code directly?

Moreover, the definition of picf in Equation (1) does not seem to correspond to the one in https://www.gtap.agecon.purdue.edu/models/setsvariables.asp (it is the total CIF price, not only the "insurance and freight" which, in my opinion, are surcharges to a FOB price).

Thirdly, the discussion of results should include some connection to the results of the paper, so far the section looks more like previous research rather than discussion, since mostly third-party studies are examined.

Finally, I think 11 references is not enough for this level of research and I recommend adding a little more studies on GTAP at least. But I would let the editor decide on this one.

Apart from methodology and discussion, I recommend one small formal change: equations (1) - (4) seem to be extracted form the TABLO language, please, format them in MS Word or TeX equation format like the GTAP webpage does (consult the link above).

Author Response

We would like to thank the comments and suggestions of the reviewer, who have greatly enriched the paper. They were all included in the new version of the paper and highlighted in red.

Point 1: I think that the methodology section should be improved to render the text more comprehensible for researchers less familiar with the GTAP CGE model, as well as the discussion of results section should be extended.

Firstly, more comments on the utility (especially EV, in the first occurrence of this abbreviation /p. 12, footnote 7/ it is not explained), three-level production function (mostly its relation to qo) and elasticities (used in the sensitivity analysis) should be provided. The term "savings prices" should be exemplified as well. Was the model computed with the help of RunGTAP? From the text it seems so but any non-standard GEMPACK computation must be documented.

If I may, I would also suggest using the terms "income and substitution effect" for the effects defined on pages 3 and 4, lines 128-139.

Response 1: More detailed comments on the utility (EV), based on Burfisher (2011), were provided in footnote 12. A more detailed explanation of the three-level production function was included in page 3 and footnotes 3 and 4, as suggested by the reviewer. The elasticities of substitution used in the sensitivity analysis were included in appendix C. The term "savings prices" was better described in footnote 5. Finally, it was explained the model was computed with the help of RunGTAP in footnote 11.

Point 2: Secondly, I have a few equations regarding the equations on pages 3 and 4. Where were they taken from? The general description of GTAP variables does not mention any of them: https://www.gtap.agecon.purdue.edu/models/setsvariables.asp Was it the Manual or the TABLO code directly?

Moreover, the definition of pcif in Equation (1) does not seem to correspond to the one in https://www.gtap.agecon.purdue.edu/models/setsvariables.asp (it is the total CIF price, not only the "insurance and freight" which, in my opinion, are surcharges to an FOB price).

Response 2: The equations were obtained from AnalyseGE, which uses TABmate and ViewHAR written by Mark Horridge and is available from the RunGTAP software as stated in footnote 6. The definition of pcif in Equation (1) was changed according to the reviewer´s comments in page 4.

Point 3: Thirdly, the discussion of results should include some connection to the results of the paper, so far the section looks more like previous research rather than discussion, since mostly third-party studies are examined.

Response 3: Two new paragraphs were included in order to address the reviewer’s comments in page 17.

Point 4: Finally, I think 11 references is not enough for this level of research and I recommend adding little more studies on GTAP at least. But I would let the editor decide on this one.

Response 4: New references dealing with the US-China trade war were included in footnote 14.

Point 5: Apart from methodology and discussion, I recommend one small formal change: equations (1) - (4) seem to be extracted from the TABLO language, please, format them in MS Word or TeX equation format like the GTAP webpage does (consult the link above).

Response 5: Equations (1) - (4) received a new format following the reviewer’s suggestion.